# Genome-Wide Characterization of Lectin Receptor Kinases in *Saccharum spontaneum* L. and Their Responses to *Stagonospora tainanensis* Infection

**DOI:** 10.3390/plants10020322

**Published:** 2021-02-08

**Authors:** Zhoutao Wang, Hui Ren, Fu Xu, Guilong Lu, Wei Cheng, Youxiong Que, Liping Xu

**Affiliations:** Key Laboratory of Sugarcane Biology and Genetic Breeding, Ministry of Agriculture and Rural Affairs, Research Centre of National Sugarcane Engineering Technology, Fujian Agriculture and Forestry University, Fuzhou 350002, China; wzt1417@126.com (Z.W.); hui224364@163.com (H.R.); xufu1219@126.com (F.X.); luguilong666@126.com (G.L.); qpchengwei@163.com (W.C.); queyouxiong@126.com (Y.Q.)

**Keywords:** lectin receptor kinases, *Saccharum spontaneum*, genome-wide characterization, sugarcane leaf blight, disease responses

## Abstract

Sugarcane is an important sugar and bioenergy ethanol crop, and the hyperploidy has led to stagnant progress in sugarcane genome decipherment, which also hindered the genome-wide analyses of versatile lectin receptor kinases (LecRKs). The published genome of *Saccharum spontaneum*, one of the two sugarcane ancestor species, enables us to study the characterization of LecRKs and their responses to sugarcane leaf blight (SLB) triggered by *Stagonospora tainanensis*. A total of 429 allelic and non-allelic LecRKs, which were classified into evolved independently three types according to signal domains and phylogeny, were identified based on the genome. Regarding those closely related LecRKs in the phylogenetic tree, their motifs and exon architectures of representative L- and G-types were similar or identical. LecRKs showed an unequal distribution on chromosomes and more G-type tandem repeats may come from the gene expansion. Comparing the differentially expressed LecRKs (DELs) in response to SLB in sugarcane hybrid and ancestor species *S. spontaneum*, we found that the DEL number in the shared gene sets was highly variable among each sugarcane accession, which indicated that the expression dynamics of LecRKs in response to SLB were quite different between hybrids and particularly between sugarcane hybrid and *S. spontaneum*. In addition, C-type LecRKs may participate in metabolic processes of plant–pathogen interaction, mainly including pathogenicity and plant resistance, indicating their putative roles in sugarcane responses to SLB infection. The present study provides a basic reference and global insight into the further study and utilization of LecRKs in plants.

## 1. Introduction

As a leading member of the membrane-anchored pattern-recognition receptors (PRRs) family, receptor kinases (RKs) play crucial roles in perceiving and transmitting diverse signals and stimuli at the cellular level in plants challenged constantly by abiotic and biotic factors [1]. In *Arabidopsis*, almost all RK members have both of the amino-terminal extracellular signal domain and the transmembrane region, except that there is only one transmembrane region in proline-rich extension like RKs [2,3]. The signals from the environments or other cells, including polypeptides, carbohydrates, steroids and microbial cell-wall components, are transmitted through signal biomolecules [2]. As a super protein family, RKs contain at least 15 subfamily members, with over 600 in *Arabidopsis* and 1000 in rice [3,4]. Depending on signal domains, each RK subfamily can specifically recognize one type of signal biomolecules as ligands [2]. Once the signal domains bind to ligands to form a complex, the intracellular protein kinase domains can be activated via transmembrane domain (TM) and continue to transmit signals to downstream proteins, such as receptor cytoplasmic kinases and MAP kinase (MAPK) cascades [2,5].

In accordance with the identities of three types of signal domains, lectin receptor kinases (LecRKs) can be divided into three categories, C-type, L-type and G-type [2,6]. However, LecRKs are determined to be plant-specific, and no homologue of LecRKs has been reported so far in human and yeast genomes [7]. The extracellular lectin domain of G-type LecRKs closely resembles the bulb-lectin proteins in humans and animals. However, since the existence of the S-locus domain is known to be involved in pollen self-incompatibility, G-types can also be defined as S-domain RKs [6]. Similarly, calcium-dependent lectin was found in the proteins of mammals, which mainly binds to various sugar moieties on the surface and mediate innate immunizations [2]. Legume lectin is originally found in the seeds of leguminous plants. Following a similar pattern as that of calcium-dependent lectin, sugars especially various complex disaccharides, are the main ligands of legume lectin [2].

As the major mediums for plant cell-to-extracellular communication, RKs involve a wide range of processes, mainly including abiotic stress response and disease resistance, as well as plant development and plant–mycorrhizal interaction [1,8]. In the zigzag model represented by Jones et al. [9], RKs can activate PAMP-triggered immunity (PTI) at phase one by recognizing microbial- or pathogen-associated molecular patterns. Combined with rapid, strong/intense effector-triggered immunity (ETI), PTI is considered to confer resistance to a broad spectrum of pathogens and constitutes the basal disease resistance of plants [9]. In view of the fact that there is no clear distinction between PTI and ETI, some effectors could also be recognized in the apoplast by extracellular receptors structurally similar to PRRs, and thus “zigzag model” was queried and a new “Spatial Invasion model” was proposed [10]. Besides, there is no doubt on the importance of PRRs in plant disease resistance in any model. Research reports on seeking resistance genes from LecRK family have increasingly appeared in recent years [11,12,13], reflecting the importance of LecRKs in plant disease-resistant breeding. An L-type *LecRK-V* accession from *Haynaldia villosa* markedly enhanced the powdery mildew resistance at both seedling and adult stages of wheat Yangmai158 [11]. LORE, belonging to a G-type protein and conferring sensitivity to lipopolysaccharide in tobacco, may increase the resistance to bacteria in crops [12]. The *Rphq2* and *Rph22* loci in cultivated barley (*Hordeum vulgare*) and wild bulbous barley, respectively, could affect the resistance of barley to leaf rust [13]. In addition, both genes encode lectin receptor kinase Hv-LecRK, and overexpression of *Hv-LecRK* in “Golden SusPtrit” barley varieties, which is susceptible to non-host leaf rust pathogen, can significantly improve the resistance of barley to leaf rust [13]. For abiotic stresses, several studies on LecRKs have also been reported [14,15,16]. For example, a LecRK from *Pisum sativum* can enhance salinity stress tolerance of transgenic tobacco by alleviating the ionic and osmotic components of salinity stress [14]. In addition, G-type RKs mediate the symbiotic interaction between the ectomycorrhizal fungus *Laccaria bicolor* and *Populus*, which provides a stepping stone for the further study of plant–mycorrhizal interaction [17]. Taking all the above into consideration, the LecRK is a multi-talented family, which is waiting for us to explore its potential and underpin crop improvement for biotic resistance and abiotic tolerance.

With the rapid development of genomics, genome-wide identification of gene families has become available. At present, the genome-wide characterization of LecRKs has been reported in several species, such as *Setaria italica* [18], *Solanum lycopersicum* [19], *Populus* [6], *Oryza sativa* and *Arabidopsis* [7]. These efforts at different plant species have revealed important information about basic structure, the evolutionary relationship among different species of the LecRK family, and also on core conservative motif involved in gene functions, and expression profiles under different adverse conditions. However, the complexity of the sugarcane genome leads to the stagnant progress in genome decipherment [20], which also hindered the genome-wide analysis of gene families in *Saccharum*. Gratifyingly, the genome sequence of one *S. spontaneum* accession named AP85-441, belonging to one of the two ancestor species of modern sugarcane hybrids (*Saccharum* spp. hybrids, Poaceae), has been published [20]. Sugarcane hybrids are derived from interspecific crosses between *S. officinarum* and *S. spontaneum*, and have approximately 10–15% chromosomes derived from *S. spontaneum* which is the most important contributor to many vital biotic resistances and abiotic tolerances [21,22].

Sugarcane, accounting for about 80% of total sugar, is cultivated in more than one hundred countries in the world. It is crucial for the sugar production and bioenergy ethanol industry. Undoubtedly, the analysis of versatile LecRKs in *S. spontaneum* represents a vital precondition to utilize them in sugarcane breeding. In this study, we analyzed in detail the evolution, domain and motif architectures, *cis*-elements and microsatellite distributions, gene localization, as well as the expression profiles of LecRKs in *S. spontaneum* and modern sugarcane hybrids in response to *Stagonospora tainanensis*, which causes SLB, a most serious fungal leaf disease-causing leaf bright and resulting in rapid leaf senescence for sugarcane [23].

## 2. Results

### 2.1. Identification and Comparison of LecRKs from Six Poaceae Species

After obtainment of LecRKs from six Poaceae species by Hmmer and Blastp, the lectin domains were extracted and built into species-specific HMM profiles. All species-specific HMM profiles from each of the six species were used for further search in corresponding species to get LecRKs as much as possible. The final search results were shown in Table 1. We identified 429 non-allelic and allelic LecRK proteins in *S. spontaneum*, including 3 C-, 160 L- and 296 G-types (Appendix A). In addition, we obtained 60, 84, 46, 55 and 56 L-type proteins from *O. sativa*, *Brachypodium distachyon*, *Zea mays*, *S. italica* and *Sorghum bicolor*, which were encoded by 57, 67, 44, 55 and 54 genes, respectively. For G-type, 93, 122, 85, 70 and 99 proteins were obtained from the five Poaceae species mentioned above, which were derived from 89, 81, 66, 68 and 86 genes, respectively. Following a similar pattern as that of *S. spontaneum*, we found that C-type proteins from the above five species were much less than those of L- and G-types, which just two, two, one, two and one proteins, respectively. In addition, the C-type proteins in each of the above five species correspond to only one coding gene.

The percentage and density of LecRK genes in each of the six Poaceae species were calculated. Among them, the percentage of LecRK genes in *S. spontaneum* was the maximum (0.51%), followed by *B. distachyon* (0.42%), and the minimum percentage observed in *Z. mays* (0.28%). Although both *S. spontaneum* and *Z. mays* belong to Panicoideae, nearly a twofold larger percentage of LecRK genes in *S. spontaneum* than that in *Z. mays*. Then, in terms of the average LecRK gene density in the genome, the *B. distachyon* possesses the maximum (0.55 Mb^−1^), followed by *O. sativa* (0.39 Mb^−1^). Although the highest percentage of LecRK genes appeared in *S. spontaneum*, its average density (0.14 Mb^−1^) in the genome was quite lower and just higher than that of *Z. mays* (0.05 Mb^−1^).

### 2.2. Basic Characteristics of LecRKs

Five typical domain architectures of three types of the LecRKs were mapped by domain analysis (Figure 1a). Only two domains of lectin and kinase existed in C- and L-type LecRKs. In G-type LecRKs, except for lectin and kinase domain, there are two other typical domains S-locus and plasminogen/apple/nematode (PAN), which improved the architectural diversity of G-type proteins. It should be noted that in G-type proteins, most of the sequences (94.7%) contain the PAN domain. Similar to other gene families, such as NLR family [24], atypical domains also can be found in LecRKs. Total 12 atypical Pfam domains were identified in 10 LecRKs using Pfam database (Appendix A). The existence of these atypical domains also enhances the diversity of the LecRK structure. Overall, the length of G-type proteins is longer than L-type, which corresponds to the number of typical domains they carry (Figure 1b).

In order to verify whether the three types of LecRKs evolved independently, 314 LecRK proteins with complete kinase domains were selected as the research objects. Because the kinase domains are more conservative than lectin domains [6], the sequence of kinase domains in each selected protein was extracted to construct the phylogenetic tree. An interesting result was presented in Figure 1c. Clearly, three types of LecRKs were distinctly segregated and have independent evolutionary branches with each other (C-, L- and G-type groups). However, there were two evolutionary branches in G-type LecRKs (G-type groups I and II), indicating relatively independent evolution between the two branches.

### 2.3. Prediction of Domain Orientation and Layout Patterns in Membrane

As shown in Figure 1a, five typical domain architectures were displayed. However, detailed domain orientation and layout patterns of LecRKs on the plasma membrane, including TM, were unknown. Here, based on the prediction results of the TM location and the orientation of the domains lectin, kinase, PAN and S-locus, all putative architectures of LecRKs domains in the membrane were analyzed (Figure 2). According to putative domain architectures, C-, L- and G-type LecRKs were categorized into 3, 22 and 47 different patterns, respectively. Although the three C-type LecRKs were allelic, their architectures may be different (Figure 2a). In 147 L-type genes (Figure 2b), almost all their architectures contained either an extracellular lectin or a membrane-embedded lectin, except patterns 5 and 8 (corresponding to 23 genes). As the most typical domain architecture, pattern 4 has the largest number, followed by patterns 11 and 14. Compared with the L-type gene, the putative layout patterns of the G-type LecRK were more diverse (47 patterns, Figure 2c), mainly because the G-type LecRK has two additional domains, i.e., PAN and S-locus. Except for nine patterns, including 10, 12, 13, 15, 16, 18, 19, 22 and 26 (corresponding to 56 genes, 21.1%), other patterns have at least one lectin located outside or inserted into the cell membrane. Several patterns, including patterns 1 and 2 in the L-type, and 1–7 in the G-type, do not contain TM, which may due to the reasons of the TM in LecRKs with these patterns having not been predicted, or having been lost in the process of evolution.

### 2.4. Motif, Domain and Exon Architectures of the Representative LecRKs

Totally 158 representative non-allelic LecRKs (52 L- and 106 G-types) that contained both complete lectin and kinase domains were selected for motif architecture analysis (Figure 3). Based on the 52 L- and 106 G-type LecRKs, respectively, the phylogenetic trees were constructed using their kinase sequences and a total of 15 highly conserved motifs were predicted (Appendix A) in the L- and G-type LecRKs, respectively. According to the phylogenetic tree, all sequences carrying motifs were rearranged.

In L-type LecRKs, except motifs 4 and 11, all other motifs were located in legume lectin (7) or kinase (6) domains. Three are four subgroups (I–IV) based on the phylogenetic tree (Figure 3a). LecRKs within the same subgroup had similar motif architectures in terms of the category and the order of motifs. Interestingly, the motif architectures among four subgroups have some obvious different characteristics, such as motifs 11 and 14 only appeared in subgroup IV; motifs 15 only appeared in subgroup I, II and IV; motif 8 appeared mainly in subgroup IV. In G-type genes, following a similar pattern as that of L-type genes, eight subgroups were divided and the genes in the same subgroup were highly similar in motif and domain architectures (Figure 3b). The differences of the motif and domain architectures among eight subgroups were clearly presented, for instance, a few of S-locus domain appeared in the subgroups I, III and VI; no motif 1 appeared in the subgroup VII; a few of motif 7 appeared in the subgroups III, IV and VI; no motifs 9 and 11 appeared in the subgroups VII and VIII; and motif 15 only appeared in the subgroups VII and VIII. In addition, we could clearly observe that, like the L-type LecRKs, most of the motifs were distributed in the domains of the bulb-lectin, PAN, S-locus and kinase (Figure 3b).

In addition, exon architectures were analyzed, and the number of exons was calculated and tagged on the behind of the LecRK sequences (Figure 3). The average number of exons contained in these genes were 2.7 and 5.0 in L- and G-type LecRKs, respectively, while the number of exons in different genes varied greatly in both types. Obviously, the exon architectures of G-type LecRKs were more complex than that of the L-type. In the G-type, 25 LecRKs contained only one exon, but two genes *LecRK-G003-2* and *LecRK-G069-1*, have more than 15 exons (15 and 21, respectively). However, in the L-type, the difference in the number of exons was smaller than that in the G-type, such as 21 LecRKs contain only one exon and the maximum number of exons was 13 (*LecRK-L065-1*). Interestingly, the exon numbers and architectures were also related to the evolutionary relationship of LecRKs. The LecRKs in subgroup III have more exons (mean: 4.7) than those in other subgroups (mean: 2.3). Similarly, all genes in the subgroups VII and VIII have more than six exons (mean: 7.9), however, most of the genes in other subgroups have less than four (mean: 3.4).

### 2.5. Cis-Element and Microsatellite Analysis

In order to explore the factors that regulate the expression of LecRKs, 115 putative *cis*-elements were predicted in the upstream promoter regions (−1500 bp) of start codon using the PlantCARE database and 236 microsatellites were screened (Appendix A). The 115 putative *cis*-elements occupy 36,780 sites, tightly covering the promoter regions of LecRK genes. The first 42 putative *cis*-elements with the maximum number of occupied sites (total 34,958, 95.0%) and covered genes more than 100 were concerned (Figure 4), which involved essential elements (red, 2), hormone responsiveness (sky blue, 9), abiotic stress (green, 3), light responsiveness (orange, 10), development (purple, 4) and other elements (black, 14). 

There is no doubt that *cis*-element is an important factor regulating gene expression. However, in addition to being an important molecular marker, microsatellite may also be a regulator of gene expression [25]. In total, 263 microsatellites were identified from 141 LecRK genes (2 C-, 34 L- and 105 G-type genes) and 57 genes contain more than one microsatellite.

### 2.6. Distribution of LecRK Genes on Chromosomes

All LecRKs (266 G-, 160 L- and three C- type genes) were mapped to eight homologous groups (Chr1-Chr8) of four members (A–D) each (Figure 5). Overall, there were significant differences in the number of LecRKs distributed on different homologous groups. Among them, Chr5 (89) and Chr2 (88) carried more than 80 LecRKs, followed by Chr7 (59), Chr8 (53), Chr1 (46), Chr3 (42), Chr4 (26) and Chr6 (26). Interestingly, different types of LecRKs also have different distribution characteristics on eight homologous groups. First, the C-type LecRKs (marked with purple) are distributed on Chr3, which contains three alleles (*C01-1*, *C01-2* and *C01-3*). For the G-type LecRKs (marked with green), its members were dominant on Chr5 (80, 30.1%), Chr7 (41, 15.4%), Chr3 (35, 13.2%) and Chr8 (34, 12.8%), respectively, accounting for 71.5% of the total. Unlike G-type LecRKs, L-type LecRKs (marked with orange) mainly distributed on Chr2 (46, 28.8%) and Chr1 (29, 18.1%). On the whole, LecRK genes tend to distribute at both ends of chromosomes, and there were fewer genes near centromere. A total of 25 duplicated gene sets involving 65 genes (47 G- and 18 L-type genes, respectively) were found in 13 chromosomes, including Chr1A (1), Chr1B (1), Chr2A (2), Chr2B (1), Chr3D (1), Chr4C (1), Chr5B (4), Chr5D (3), Chr6C (1), Chr7B (2), Chr7C (3), Chr7D (1) and Chr8C (4), in which just six sets of genes belonged to L-type (Figure 5).

### 2.7. Expression Pattern in Response to Leaf Blight in Sugarcane Hybrids ROC22 and FN12-047

At different stages of disease development, the morphology of disease lesions on the leaves of resistant- and susceptible-sugarcane lines was basically the same, although there were significant differences in the number of lesions. At the medium to latestage, the lesions appeared to be obviously different between two lines: the lesions almost covered the whole leaf in the susceptible line FN12-047, while only a few lesions appeared in the resistant line ROC22 (Figure 6a). On the other hand, the spores of the pathogen *S. tainanensis* can be observed on the disease lesions in both susceptible and resistant sugarcane lines. (Figure 6b) A total of 61 DELs (40 G- and 21 L-type genes, FDR < 0.05, |log(FoldChange)| > 1.0) were detected by DESeq2 in transcription reprogramming data, which were derived from SLB-resistant sugarcane accession ROC22 and -susceptible sugarcane accession FN12-047 infected by *S. tainanensis* in early and medium to late stages of SLB development (Figure 6c). In both varieties, the number of DELs was obviously less in the early stage, but increased significantly in the medium to late stage. In general, most of the genes were up-regulated, especially for resistant ROC22 in the medium to late-stage, in which all genes were up-regulated. Only less than half (24) of the DELs were shared (marked with blue squares) between the two hybrids. However, the expression trend (down-regulated or up-regulated) of each shared DEL was consistent between the hybrids.

### 2.8. The Comparison of the Expression Pattern in Response to Leaf Blight in Three Sugarcane Accessions

Although the expression dynamics of LecRKs in response to SLB in sugarcane hybrid transcriptomes having been detected, the response of LecRKs in *S. spontaneum* may be divergent due to the obvious difference in genetic background between *S.* spp. hybrids and *S. spontaneum*. Therefore, we carried out a special study on the expression pattern of LecRKs in *S. spontaneum* infected by *S. tainanensis* at the same time. A total of 35 DELs (19 G- and 16 L-type genes) were detected in the transcriptome of susceptible *S. spontaneum* SES208 infected by *S. tainanensis* (Figure 7a). Compared with the early stage of SLB in which only eight DELs obtained, more LecRK genes (32) showed to be changed significantly in their expression in the medium to late stage of disease, in which the number of up-regulated genes (26) was obviously higher than that of the down-regulated (6). However, no obvious difference was observed in the total number of DELs among the two hybrids (42 for ROC22 and 43 for FN12-047) and SES208 (35). Based on the Venn diagram (Figure 7b) of the DEL sets derived from three accessions, only six genes (gene set I) shared by them. In addition, ROC22 shared 24 DELs (gene sets I and III) with FN12-047 and shared nine DELs (gene sets I and IV) with SES208. Similarly, FN12-047 shared more DELs with ROC22 than with SES208. Each accession has its own unique DEL set, of which SES208 has the most.

We did not find that the C-type genes were differentially expressed in the leaves during SLB development, either in two hybrids or in *S. spontaneum.* However, there is a definite fact that the C-type genes, especially the allelic gene *C01-1*, were highly expressed in all samples of both hybrids and *S. spontaneum* (Figure 7c). In hybrids ROC22 and FN12-047, the expression of the allelic gene *C01-2* was lower than those of the other two alleles, and same for the *C01-3* gene in *S. spontaneum*.

### 2.9. qRT-PCR Expression Analysis Proved That the Transcriptome Data Were Reliable

Gene expression analysis based only on the transcriptome may lead to the deviation, which mainly due to the possible sequencing bias and the inapplicability of analysis methods. Considering that the LecRK family genes were identified from the genome of *S. spontaneum* SES208, the accession of SES208 was used to verify the consistency of the gene expression between the transcriptome analysis and the qRT-PCR detection. The relative expression levels of the total 20 genes were verified in 12 samples of SES208 (Figure 7d). Among these 20 genes, four were non differentially expressed genes and 16 were differentially expressed genes in SES208. Surprisingly, the results of qPCR and the transcriptome analysis were consistent, especially gene expression trends in different disease stages. Besides, four no differentially expressed (non-DELs), *C01-1*, *G005-2, G020-1* and *G018-1*, also showed almost the same expression level in the qPCR validation. However, for the fold changes, the results of the qRT-PCR and transcriptome analysis were not absolutely consistent, and slight error exists. Thus, the reliability of transcriptome occupied here is supported by the results of qRT-PCR.

## 3. Discussion

Several gene families have been identified and studied in *Saccharum* based on transcriptome or genome scaffolds [26,27,28]. However, compared with the transcriptome or scaffolds, identification and analysis of gene families from chromosome-scale assembly can provide more information, for instance, the gene distribution in chromosome and upstream *cis*-elements in promoters. Moreover, transcriptome merely contains transcripts that emerged in a specific spatio-temporal expression, which results in incomplete identification of all members of the gene family, especially for their complete architectures. In the current study, based on the assembled chromosomes of *S. spontaneum*, a crucial gene family in plants stated above, LecRKs, was identified and analyzed meticulously.

### 3.1. Expansion and Evolution of LecRKs in S. Spontaneum

In order to objectively compare the quantity and density of LecRK genes in the six Poaceae species, an identical strategy was used to scan LecRK members from the six Poaceae species. Finally, 429, 145, 145, 105, 118 and 138 LecRK genes were identified from *S. spontaneum*, *O. sativa*, *B. distachyon*, *Z. mays*, *S. italica* and *S. bicolor*. Previous studies demonstrated that the identified LecRK numbers in *O. sativa* and *S. italica* are consistent with our results [7,18]. Although the LecRK gene percentage of *S. spontaneum* (0.51%) was the highest compared with the other five Poaceae species, its density (Mb^−1^) was low (0.14). From LecRKs stock of monoploid reference sequence of six Poaceae species, *B. distachyon*, a wild annual grass endemic to the Mediterranean and Middle East [29], is undoubtedly an excellent LecRK resource repository. In *Arabidopsis*, *Solanum lycopersicum* and *Populus*, a total of 75, 93 and 231 LecRK genes were identified, respectively [6,7]. For the reason of LecRKs involving in various abiotic stress response and disease resistance in plants, the large number of LecRKs may help *Populus* maintain a longer life cycle [6]. Expansion of LecRKs in *Populus* is obvious, which is indicated by the number of G-type LecRKs, since the sum is over three times that of L-type LecRKs [6]. The lower expansion of G-type genes has also been observed in *S. spontaneum* and in the other five Poaceae species (Table 1), in spite of the expansion level of G-type genes in *S. spontaneum* (G-types/L-types = 1.67) was slightly higher than those in the other five Poaceae species. Contrarily, in *Arabidopsis*, the L-type other than G-type genes were expanded [7,30]. In *S. spontaneum*, the expansion of G-type genes was mainly due to the tandem replication. After all, of the 25 duplicated gene sets detected, 18 (72.0%) belonged to the G-type (Figure 5). Interestingly, though the C-type LecRKs can be found in a large number of mammalian systems, only one C-type LecRK (or one group allelic C-type LecRKs) can be found in the six Poaceae species and many other plants, such as *Arabidopsis*, *S. lycopersicum* and *Populus* [6,7,19,31].

The evolutionary relationship of three types (C-, L- and G-types) of genes was constructed using a complete kinase domain from 314 selected proteins. The fact indicates that the three types of genes evolved independently (Figure 1c), which was also found in the LecRK family of *Populus* [6]. Similarly, in *S. lycopersicum*, the unrooted neighbor-joining distance tree demonstrates that the evolution of the three types of the LecRK family is also strictly independent [19]. Based on this, we conclude that these three independently evolved gene types may have appeared in plants before the formation of species *S. spontaneum*, *S. lycopersicum* and *Populus*.

### 3.2. Domain and Motif Architectures

Detailed motif and exon architectures of 158 representative non-allelic LecRKs and domain layout patterns of all LecRKs were analyzed (Figure 2 and Figure 3). Following a similar pattern as that of other species, such as rice, *Arabidopsis*, as well as *S. italica*, S-locus region and PAN motifs existed only in the G-type LecRKs [7,18]. Because G-type LecRKs contain two additional domains, more domain architectures (47) were predicted in G-type genes (Figure 2). Although in most cases, lectin or other signal domains (such as leucine-rich repeat) are located outside the cell to sense environmental stimuli, the kinase is located in intracellular to transmit signals to activate conserved and convergent Ca^2+^ signaling and MAPK cascades to regulate the downstream process [32]. However, many domain architectures indicate that the orientations of lectin and kinase may be interchanged (Figure 2). Even, lectin and kinase are both located inside and outside the plasma membrane. These unusual architectures were also reported in other species, such as *Poplus* [6]. The diversity of domain architectures is very important for the ability of plants to sense external signals, and some abnormal structures may have unexpected functions. Diverse ligands or molecular patterns may be derived from bacteria, fungi, oomycetes, viruses, nematodes, parasitic plants and host plants and so on [33], which also require plants to have sufficient PRR diversities to cope with the challenges.

We predicted 15 most conserved motifs from 52 L- and 106 G-type representative LecRKs, respectively. Except for motifs 4 and 11 of the L-type, almost all motifs are distributed in the domains (Figure 3), which indicates that the domains of LecRKs are highly conserved. Compared with the highly conserved kinase domains, lectin domains are less conserved, among which just five motifs (three G- and two L-types) were conserved in rice and *Arabidopsis* [7]. However, we noted that four of the five motifs mentioned are also conserved in *S. spontaneum* (motifs 6, 10 and 13 in G-type; motif 5 in L-type). Low conservation is conducive to the diversity of functions and recognizable signals of the lectin domain [7]. However, doubtlessly, the highly conserved motifs play a crucial role in maintaining the basic function and structure of the lectin domain.

Motif and exon distributions display that the most closely related members in the phylogenetic tree have similar or even the same motif and exon architectures, which points to the existence of functional similarities among LecRKs (Figure 2 and Figure 3).

### 3.3. LecRK-Expression Analysis Response to SLB

We analyzed the expression of all genes from the LecRK family during the development of SLB. A question may make us confused that more than 35 genes are differentially expressed in each sugarcane accession. Are all these genes related to leaf blight resistance? A previous study reported that multiple *Arabidopsis* L-type LecRK T-DNA insertion lines, infected by *Phytophthora* pathogens, showed a significantly higher disease severity index, which may indicate that this is not an unreasonable phenomenon [34]. Comparing the DELs in different sugarcane accessions, several gene sets (I–VI) with specific meanings were identified. The DEL number in the shared gene sets was highly variable, which indicates that the expression dynamics of LecRKs in response to SLB are quite different among different sugarcane accessions. In particular, the difference between hybrids and ancestor species *S. spontaneum* was significantly greater than that between two hybrids, which may due to the much different genetic background between hybrids and *S. spontaneum*. We know that sugarcane hybrids only possess 10–15% chromosomes of *S. spontaneum*, and the others are from *S. officinarum* (about 80%) or recombinant chromosome. A total of 85 DELs were detected in three sugarcane accessions. This provides a range to study the key LecRKs responding to SLB. In particular, the DELs shared by three accessions (gene set I) need to be further cloned and sequenced to analyze their sequence variation among three accessions.

There are also several non-DELs with high expression level that we tend to define as development-related receptors, including three C-type alleles. In particular, the expression level of *C01-1* in leaves was significantly higher than that of the other two alleles in both hybrids and *S. spontaneum*. At present, there is no specific report on the role of C-type gene in plants. In mammals, C-type lectin motifs are found in proteins that mediate innate immune responses [35,36], which leads us to consider whether these proteins also function in pathogen detection and response in plants [2]. However, the current study makes us more inclined to speculate that C-type may participate in a key or basic metabolic process in plants because they are expressed in three sugarcane accessions regardless of pathogen infection.

Here in our study, there are obvious differences in the expression levels of a number ofLecRKs between susceptible and resistant *Saccharum* accessions, which suggest that the LecRKs might indeed respond to the infection of *S. tainanensis*. We are still not sure which gene plays a key role in the resistance of sugarcane to SLB, and thus more biological approaches such as transient expression and knockout of genes [37,38] should be used to investigate the most critical resistance response of these candidate LecRKs.

## 4. Materials and Methods

### 4.1. S. spontaneum and Other Five Poaceae Species Genome Resources

The genome assembly (v4.1) of the haploid *S. spontaneum* (from SES208), AP85-441, containing eight homologous groups of four members each, as well as the genome annotation (82,773 non-allelic and allelic genes annotated), were downloaded from the following link: http://www.life.illinois.edu/ming/downloads/Spontaneum_genome/ [20]. Genome sequences and gene annotations of other five Poaceae species *O. sativa* (Japonica Group, IRGSP-1.0), *B. distachyon* (Brachypodium_distachyon_v3.0), *Z. mays* (B73 RefGen_v4), *S. italica* (Setaria_italica_v2.0) and *S. bicolor* (Sorghum_bicolor_NCBIv3) were downloaded from EnsemblPlants database with the link: http://plants.ensembl.org/species.html. The six Poaceae species can be divided into three subfamilies of Poaceae, the Ehrhartoideae (*O. sativa*), Pooideae (*B. distachyon*) and Panicoideae (*S. spontaneum*, *Z. mays*, *S. italica* and *S. bicolor*) [39].

### 4.2. LecRKs Search in S. Spontaneum and Other Five Poaceae Species

Vaid et al. identified dozens of L- and G-type LecRKs in *Arabidopsis*, while only one C-type LecRK was identified; there is also only one C-type LecRK in the rice genome, which far less than L- and G-type LecRKs [7]. Therefore, we need to develop different identification strategies to identify different types of LecRKs. Our search strategies are based on Hmmer v3 using the Hidden Markov Model (HMM). For the identification of L- and G-type, the amino acid sequences of the conserved N-terminal lectin domain of total 42 L- and 32 G-type genes from *Arabidopsis* were extracted directly by CDD tool from NCBI [40] with E-value < 1 × 10^−3^ and excel function MID, and then built the raw L-type.hmm and G-type.hmm HMM profiles after alignment using clustalw. Using the HMM profiles L-type.hmm and G-type.hmm, we initially searched for dozens of L- and G-type LecRKs from *S. spontaneum*. In order to search all LecRKs as much as possible, we extracted lectin domains from L- and G-type LecRKs from *S. spontaneum* again and built the *S. spontaneum*-specific SsL-type.hmm and SsG-type.hmm profiles in accordance with the above methods. Finally, the *S. spontaneum*-specific SsL-type.hmm and SsG-type.hmm profiles were used to search for *S. spontaneum* genome again, and all searched proteins were filtered by CDD and Pfam database with E-value < 1 × 10^−5^. The above strategy for identifying L- and G-type LecRKs was also used in the other five Poaceae species.

Because both *Arabidopsis* and rice contain one C-type LecRK, we cannot directly build C-type HMM profile like L- and G-types. Therefore, the conserved N-terminal lectin sequence of *Arabidopsis* was firstly used as a query locus and Blastp was used to initially search the genomes of six species. Then, all lectin domains of C-type LecRKs from six Poaceae species searched by Blastp were built into the C-type.hmm profile, and the further searches in the genomes of six species were implemented using C-type.hmm to find the possibly missed C-type LecRKs.

### 4.3. Sequence Analysis

In order to verify whether the three types of genes evolved independently, a total of 314 proteins with the complete kinase domain were selected as the research objects. Because the kinase domains are more conservative than that of the lectins [6], the kinase domain of each selected protein was extracted as the sequence to construct the phylogenetic tree. In addition, considering that the differences of the N-terminal lectin domains among different types of LecRKs make the alignments of this region ambiguous, thus lectin sequences are not ideal choices for the construction of the phylogenetic tree.

The multiple alignments of 314 kinase sequences was performed by MEGA X v10.0.5 using the clustalw algorithm with default parameter settings. The conserved functional motif prediction of 168 full-length representatives LecRKs was implemented using MEME suit [41] with the following parameter sets: the distribution of motifs (-mod), anr; the maximum number of motifs to find (-nmotifs), 25; and the minimum motif width (-minw) and maximum motif width (-maxw) of each motif, 10 and 50 residues. To find the regulatory elements of LecRKs, the upstream sequences (1500 bp) of the 429 LecRKs were retrieved and then submitted to the web tool PlantCARE (http://bioinformatics.psb.ugent.be/webtools/plantcare/html/) [42]. In addition, a web tool TMHMM Server v2.0 (http://www.cbs.dtu.dk/services/TMHMM/#opennewwindow) was available to predict the TM of each gene.

### 4.4. Chromosomal Distribution and Tandem Duplication of LecRKs

All LecRK genes (total 429) were mapped to eight homologous groups of four members each of *S. spontaneum* based on physical location information extracted from the gff3 annotation file of *S. spontaneum* genome using a web tool MG2C v2.1 (http://mg2c.iask.in/mg2c_v2.0/). In order to clearly display the distributions of the LecRK genes, characters “LecRK” were truncated in gene names. The number of the gene was tagged at the top of each group of chromosomes, and C-, L- and G-type genes were marked by different colors. When two genes belonged to the same gene type with sequence distances < 200 kb, alignment length ≥ 70% (compared to the longer genes) and identity ≥ 70% family, they were defined as tandemly duplicated genes.

### 4.5. Plant Materials and Treatments

Sugarcane leaf blight, caused by the pathogenic fungus *S. tainanensis* [23], is one of the most serious biotic constraints in sugarcane cultivation. The expression profiles of the LecRK genes at different development stages of *S. tainanensi* infection in *S. spontaneum* and sugarcane hybrids, are our interest. For *S. spontaneum* accession SES208, more than 50 plants were cultivated in field under the same environmental conditions in Fuzhou, China. Two different disease development stages were defined: for early stage of infection, most lesions or spots were pale yellow; for medium to late stage of infection, most lesions were reddish-brown, bright red or brown together with part of necrotic tissue. Four biological replicates were performed for each infection stage and for control (healthy), and thus total 12 samples were obtained. For each infected replicate, more than 100 lesions clipped from random leaves and random plants were pooled. Similarly, control samples were also randomly collected from different leaves of different plants.

The expression pattern of the LecRKs in response to *S. tainanensis* infection in *Saccharum* spp. hybrids with different SLB resistance was another focus in this study. The experimental data on the interaction between two sugarcane hybrids and *S. tainanensis* were used for the analysis of the expression patterns. In order to highlight differences in SLB resistance and reduce the differences caused by a genetic background, one susceptible offspring accession FN12-047, derived from the cross combination of *S. tainanensis*-resistant ROC22 and -susceptible YT93-159, together with its resistant male parent ROC22 were used to investigate the expression patterns of LecRKs in response to *S. tainanensis* infection. Both sugarcane hybrid accessions were cultivated in the field next to *S. spontaneum* SES208 under the same environmental conditions. For each sugarcane accession, two plants with similar growth vigor and disease severity were selected for sampling, and the leaf located at the same leaf position was collected, pooled together for one replicate before RNA extraction, and finally subjected to RNA-sequencing. As with *S. spontaneum*, samples from both early and medium to late stages were collected. The healthy leaves at same leaf position without any disease symptom were selected as the control. Three biological replicates were performed for each hybrid and a total of 18 samples were obtained, nine for each accession.

### 4.6. Identification of Differentially Expressed LecRKs

As an efficient and time-saving transcriptome reads alignment tool, STAR was used as the aligner in current study [43,44]. Because *S. spontaneum* is a polyploid species, STAR software has an advantage in this study due to its higher output of unique alignment [44]. For multiple alignment of a read, just one alignment with the highest score was retained to improve the accuracy of quantitative expression among alleles. Then, the tool of Featurecounts was used to quantify the gene expression [45]. DESeq2, providing the most accurate differential analysis, has been confirmed by Sme et al. [44]. For *S. spontaneum*- and hybrids-*S. tainanensis* transcriptome data, each sample has four and three biological repeats, respectively, and DESeq2 was applied as the software for the analysis of the differentially expressed genes. All differentially expressed genes need to meet the following screening criteria: adjusted *p*-value < 0.05 and |log2(FoldChange)| > 1.

### 4.7. Quantitative Real-Time PCR

Twenty LecRK genes including four non-DELs and sisteen DELs were randomly selected to verify the accuracy of the SES208 transcriptional data by qRT-PCR. The effective open reading frames (ORFs) of the twenty selected genes were predicted by the online web tool NBCI ORFfinder, which is necessary for the primer design. Then, the primers corresponding to the selected genes for qRT-PCR analysis were designed by Primer Premier 6 (Appendix A). The qRT-PCR validation strategy and process have been described in detail by Huang et al. [46], and the reference gene for qRT-PCR was glyceraldehyde-3-phosphate dehydrogenase (*GAPDH*) according to the research of Iskandar et al. [47].

## Figures and Tables

**Figure 1 plants-10-00322-f001:**
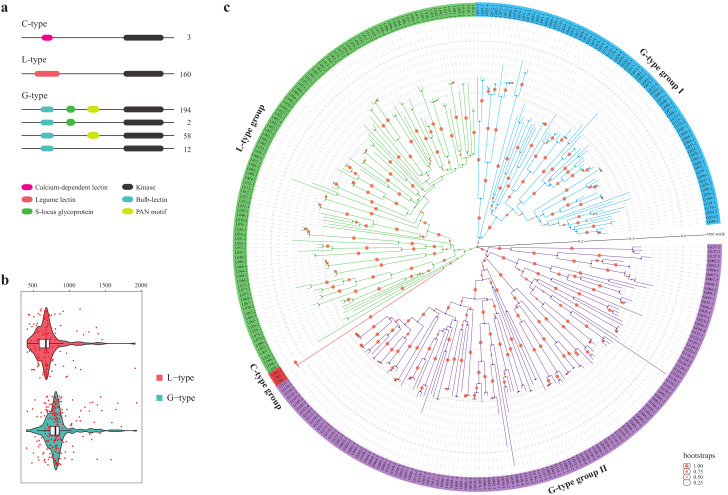
Basic characteristic analysis of 429 LecRKs. (**a**) Typical domain architectures of C-, L- and G- type LecRKs. Protein number of each architecture was presented behind the sequence. Six main domains, including calcium-dependent lectin (C-lectin), legume-lectin (L-lectin), bulb-lectin (G-lectin), kinase, plasminogen/apple/nematode (PAN) and S-locus, were marked concurrently in different colors; (**b**) The number of amino acids predicted for three types of LecRKs. (**c**) Phylogenetic tree analysis of 314 LecRKs with complete kinase domains indicates three types of LecRK proteins segregated distinctly with independent evolutionary branches.

**Figure 2 plants-10-00322-f002:**
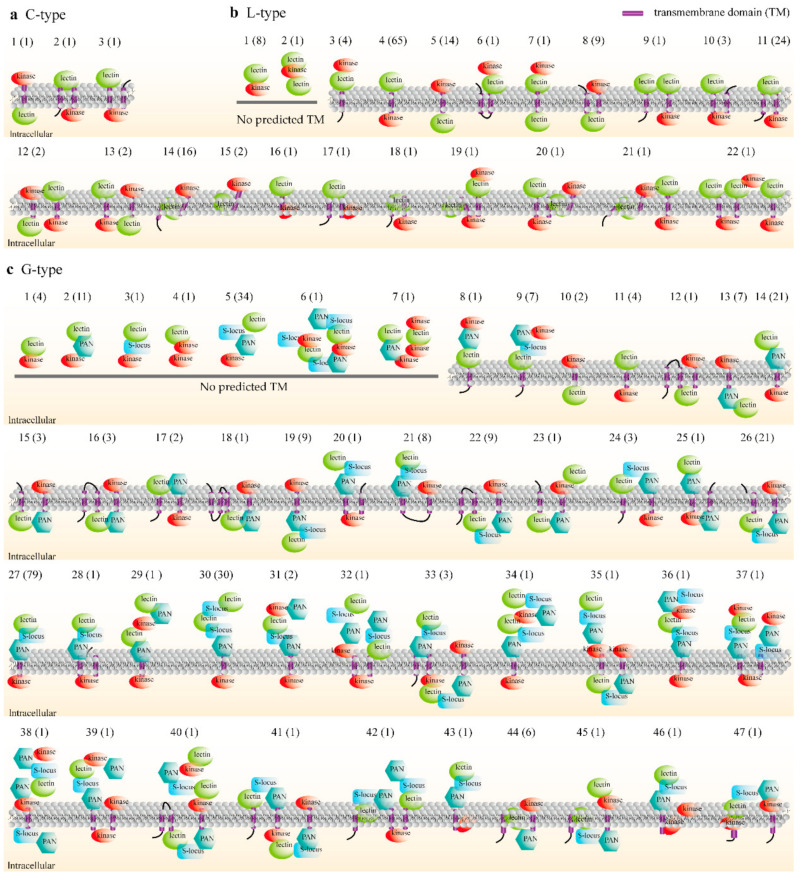
Domain architectures of LecRKs. Based on the prediction results of the transmembrane domain and the orientation of the domains lectin, kinase, PAN and S-locus, LecRKs were categorized into several different patterns. (**a**) The C-type LecRKs were categorized into three patterns. (**b**) The L-type LecRKs were categorized into 22 patterns. (**c**) The G-type LecRKs were categorized into 47 patterns.

**Figure 3 plants-10-00322-f003:**
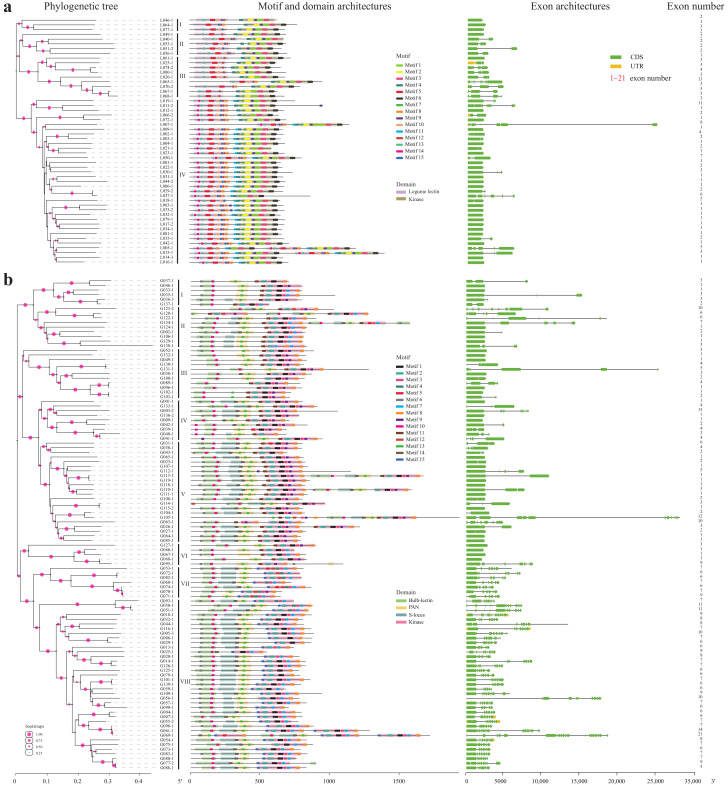
Conserved motif, domain and exon architectures of representative 52 L- and 106 G-type LecRKs. (**a**) Motif, domain and exon architectures of 52 representative L-type LecRKs. (**b**) Motif, domain and exon architectures of 106 representative G-type LecRKs. Exon number was tagged on the behind of each exon architecture.

**Figure 4 plants-10-00322-f004:**
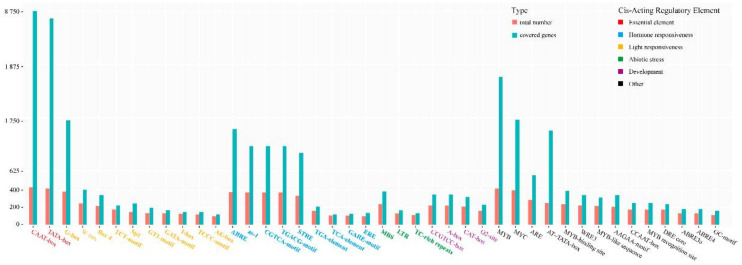
The first 42 predicted *cis*-elements with the maximum number of occupied sites and covered genes more than 100. Total number represents the sum of a *cis*-element appeared in all LecRK genes; Covered genes represent the number of genes with at least one corresponding *cis*-element.

**Figure 5 plants-10-00322-f005:**
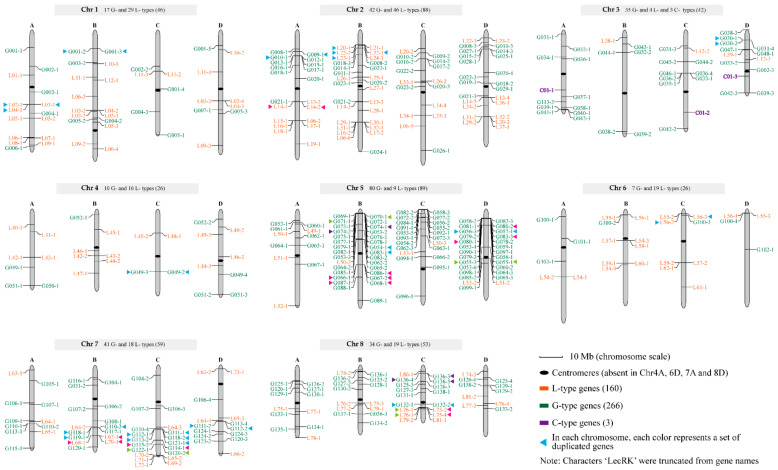
Physical distribution of LecRK genes on *Saccharum spontaneum* chromosomes. All LecRK genes are mapped to eight homologous groups (Chr1–Chr8) of four members (A–D) each. Green, orange and purple gene names on chromosomes represent G-, L- and C-type genes, respectively. Characters “LecRK” were deleted from each gene name in the figure to save space. The number of C-, L- and G- type genes are marked on top of each homologous group. In each chromosome, each color represents a set of duplicated genes and total of 25 duplicated gene sets involving 65 genes were found in 13 chromosomes.

**Figure 6 plants-10-00322-f006:**
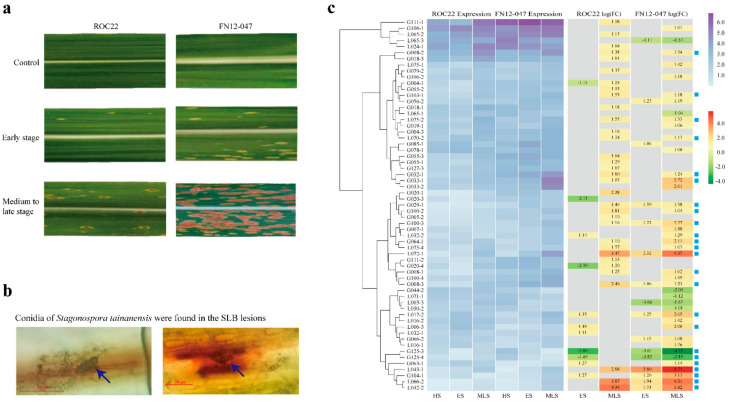
Morphology of disease lesions of sugarcane leaf blight (SLB) and expression dynamics of LecRK gene in hybrids in response to SLB. (**a**) Morphology of disease lesions of SLB in early and medium to late disease stages of resistant ROC22 and susceptible FN12-047. (**b**) The distribution of conidia of *Stagonospora tainanensis* on the surface of disease lesions of SLB. (**c**) Heatmap of differentially expressed profiles and fold change (display as log (FC)) of LecRK genes responding to SLB in healthy, early and medium to late stages (abbreviated HS, ES and MLS, respectively). Genes marked with blue squares represent shared differentially expressed LecRKs between sugarcane hybrids resistant ROC22 and susceptible FN12-047. Characters “LecRK” were deleted from each gene name in the figure to save space.

**Figure 7 plants-10-00322-f007:**
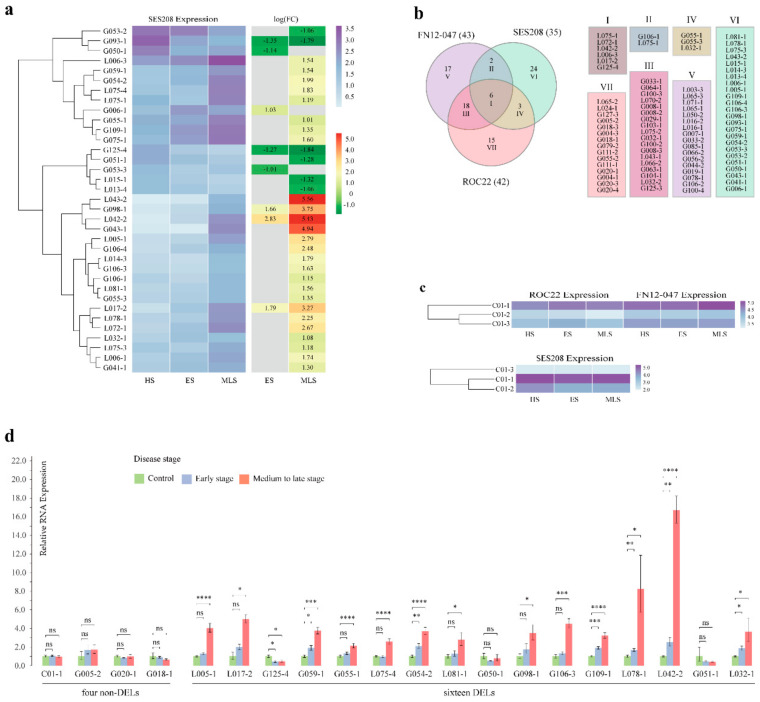
The comparisons of the expression patterns of LecRK family in response to sugarcane leaf blight (SLB) in *Saccharum spontaneum* SES208 and two hybrids ROC22 and FN12-047. (**a**) Heatmap of differentially expressed profiles and foldchange (display as log(FoldChange)) of the LecRK genes responding to SLB in the healthy, early and medium to late stages (abbreviated HS, ES and MLS, respectively) in *S. spontaneum* SES208. (**b**) Venn results of the differentially expressed gene sets of three sugarcane accessions. (**c**) Heatmap of the non-differentially expressed profiles of C-type LecRK genes. (**d**) Based on the RNA samples of SES208, the relative expression levels of a total of 20 LecRK genes were verified by quantitative real-time PCR (qRT-PCR). Among them, most except four are differentially expressed LecRKs in SES208. Significant differences are indicated by asterisks (*, *p* < 0.05; **, *p* < 0.01; ***, *p* < 0.001; ****, *p* < 0.0001) and “ns” represents no significant difference. Characters “LecRK” were deleted from each gene name in Figure 7 to save space.

**Table 1 plants-10-00322-t001:** Number and characteristics of lectin receptor kinase (LecRK) proteins/genes in the genomes of six different Poaceae species.

Content	*Saccharum spontaneum*	*Oryza sativa*	*Brachypodium* *distachyon*	*Zea mays*	*Setaria italica*	*Sorghum* *bicolor*
C-type	3/3	2/1	2/1	1/1	2/1	1/1
L-type	160/160	60/57	82/65	45/43	54/54	56/54
G-type	266/266	91/87	120/79	80/61	64/63	96/83
G-type/L-type	1.67	1.53	13.2	1.42	1.17	1.53
Total proteins	429	153	204	126	120	153
Total coding genes	429	145	145	105	118	138
# of coding genes in the genome	83,826	35,825	35,125	39,591	35,831	34,118
Percentage of LecRK genes (%)	0.51	0.41	0.42	0.28	0.35	0.41
Genome size (Mb)	3,133.3	375.0	271.2	2,135.1	405.7	708.7
Average LecRK gene density (Mb^−1^)	0.14	0.39	0.55	0.05	0.31	0.20

## Data Availability

Not applicable.

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
