# Peer review of "Genome-Wide Characterization of Lectin Receptor Kinases in Saccharum spontaneum L. and Their Responses to Stagonospora tainanensis Infection"

_plants, 2021, doi:10.3390/plants10020322_

Round 1
Reviewer 1 Report
The study by Wang et al. represents a detailed description of Lectin Receptor Kinases (LecRKs) in Saccharum spontaneum, considered an ancestor of S. officinarum and hybrids of this species that comprise the major commercial sugar cane cultivars used for sugar production worldwide. Basically, the study comprises an in silico analysis of the 429 allelic and non-allelic LecRKs identified. These were divided into their three main groups (calcium-dependent (C-lectin), legume- (L-lectin) and bulb-lectins (G-lectins). Data sets describing their main transmembrane domain arrangements, conserved motif, domain and exon distribution, type and abundance of cis-elements present in their respective promoters and chromosome localization were also included. A selection of these LecRKs was further used to determine their expression levels in response to the fungal pathogen Stagonospora tainanensis in resistant and susceptible cultivars, respectively.
Despite the massive amount of data presented, the study is descriptive in nature. It offers basically no information regarding the biological function that these particular groups of genes might have in Saccharum spp., except haphazard comments suggesting that, similar to LecRKs in other species, they might be involved in signaling pathways associated with the regulation of plant development and/ or responses to (a)biotic stress conditions. In this respect, experimentation designed to determine their participation in the regulation of carbon fixation and carbon allocation processes leading to increased sugar contents would have been a very welcome contribution. The authors do mention that some of these LecRKs are differentially expressed in resistant and susceptible Saccharum spontaneum hybrids during different stages of the infective process by the fungal pathogen S. tainanensis leading to leaf blight. However, considering the complexity of plant pathogen resistance mechanisms, there is still much to be done to define how, and to what extent, these genes contribute to the sugar cane resistance against this fungal pathogen.
Also needed are images of the infection process in the resistant in susceptible sugar cane hybrids. Moreover, the quality of all the images included in the manuscript´s figures was very low-grade. The resolution of the images must be greatly improved to correct their blurriness, thereby permitting the comprehension of the substantial and detailed information provided in each figure.
Finally, the level of the written English used in this manuscript is extremely poor. The text is hard to follow, becoming incomprehensible in several passages. It should be re-written.
Reviewer 2 Report
This manuscript was analyzed in detail the evolution, domain and motif architectures, cis-elements and microsatellite distributions, gene localization, as well as the expression profiles of LecRKs in S. spontaneum and modern sugarcane hybrids in response to Stagonospora tainanensis. In manuscript, various analyzes of LecRKs were attempted using S. spontaneum and other five Poaceae Species Genome Resources.
Line 276-295: Explaining gene expression analysis by
analyzing only the transcriptome can lead to too many errors. Therefore,
transcriptome analysis and RT-PCR analysis are usually presented together to
perform validation experiments. Therefore, for
35 DELs, RT-PCR analysis is considered essential
Round 2
Reviewer 1 Report
I have perused the revised version of MS titled ““Genome-wide Characterization of Lectin Receptor Kinases in Saccharum spontaneum L. and their Responses to Stagonospora tainanensis Infection” (plants-1043325)”. I still have several observations regarding the response to the concerns I raised about certain aspects of this study. These are the following:
- Although the level of the written English was improved, the MS still requires further refinement to be acceptable for publication in “Plants”.
- The study remains mostly a rather difficult to digest collection of abundant bioinformatics data. Apart from an attempt to connect these lectin kinase receptor´s to a biological function by relating transcript abundance with resistance/susceptibility to a fungal pathogen, its nature is essentially a descriptive one. Clear differences in expression levels of a number of these LRK in susceptible and resistance Saccharum accessions suggest tha LecRK´s might indeed be responsible for the difference observed. However, it remains unknown which of these LecRK genes are key to the resistance response observed and which are most probably a carry-over of LecRKs that were generated during the co-evolution of Saccharum plants and this particular pathogen? Or is the resistance observed the result of the sum of all LecRK genes that have accumulated as a result of this process, similarly to what has been proposed to occur in undomesticated solanaceous plants showing R gene-related resistance to infection by Phythophthora oomycete pathogens, for example potato (refer to Elnahal ASM, Li J, Wang X, Zhou C, Wen G, Wang J, Lindqvist-Kreuze H, Meng Y and Shan W (2020) Identification of Natural Resistance Mediated by Recognition of Phytophthora infestans Effector Gene Avr3aEM in Potato. Front. Plant Sci. 11: 919).
In this respect, images showing differences in disease severity between the resistant and susceptible accessions are still missing. Also a description of the possible experimental strategies that could be employed to define the role of these genes in development/ physiology (e.g, C allocation) of Saccahrum plants (or other members of the Poaceae) was lacking.
- The resolution of most of the images included in this MS is still insufficient. The text in many sectors of these figures, e.g., inside the surrounding circle, or colored sectors, of Figure 1, remains unintelligible.
Reviewer 2 Report
This paper analyzed in detail the expression profile of LecRKs in S. spontaneum and modern sugarcane hybrids in response to Stagonospora tainanensis, as well as evolution, domain and motif architecture, cis element and microsatellite distribution, and gene localization. All the points pointed out in the manuscript have been corrected, and it is judged that it can be published in the current state.